# Dilution of specialist pathogens drives productivity benefits from diversity in plant mixtures

Guangzhou Wang ◉[1,2] ✉, Haley M. Burrill ◉[2,3,4], Laura Y. Podzikowski ◉[2,3], Maarten B. Eppinga[5], Fusuo Zhang ◉[1], Junling Zhang[1], Peggy A. Schultz[2,6] & James D. Bever ◉[2,3] ✉

Productivity benefits from diversity can arise when compatible pathogen hosts are buffered by unrelated neighbors, diluting pathogen impacts. However, the generality of pathogen dilution has been controversial and rarely tested within biodiversity manipulations. Here, we test whether soil pathogen dilution generates diversity-productivity relationships using a field biodiversity-manipulation experiment, greenhouse assays, and feedback modeling. We find that the accumulation of specialist pathogens in monocultures decreases host plant yields and that pathogen dilution predicts plant productivity gains derived from diversity. Pathogen specialization predicts the strength of the negative feedback between plant species in greenhouse assays. These feedbacks significantly predict the overyielding measured in the field the following year. This relationship strengthens when accounting for the expected dilution of pathogens in mixtures. Using a feedback model, we corroborate that pathogen dilution drives overyielding. Combined empirical and theoretical evidence indicate that specialist pathogen dilution generates overyielding and suggests that the risk of losing productivity benefits from diversity may be highest where environmental change decouples plant-microbe interactions.

Diverse plant communities are consistently more productive and support other desirable ecosystem functions[1,2]. Though, productivity gains from diversity are commonly attributed to increased resource partitioning[3,4], direct evidence of resource partitioning operating in plant biodiversity-manipulation experiments has been limited[5–7]. This limited success could reflect inadequate measures of resource partitioning, or could arise from the importance of other unmeasured mechanisms. Pathogen dilution is a prominent alternative explanation, hypothesizing productivity benefits from diversity arise when compatible pathogen hosts are buffered by unrelated neighbors, diluting adverse host impacts (e.g., disease incidence, inhibited growth)[8–10]. To date, studies of pathogen dilution have focused heavily on wildlife and human diseases[11]. While meta-analyses indicate biodiversity decreases disease prevalence[8,9], the generality of pathogen dilution across systems has been controversial[12,13]. Studies of pathogen dilution in plant communities have been less common[14] and direct evidence for pathogen dilution contributing to plant productivity and ecosystem functioning remains sparse.

Testing the importance of pathogen dilution in plant communities is challenging given the high taxonomic diversity of pathogens and difficulties measuring their abundance. Conclusions about pathogen importance, for example, may depend on which of many

[1]State Key Laboratory of Nutrient Use and Management (SKL-NUM), College of Resources and Environmental Sciences, National Academy of Agriculture Green Development, China Agricultural University, 100193 Beijing, People's Republic of China. [2]Kansas Biological Survey and Center for Ecological Research, University of Kansas, Lawrence, KS 66045, USA. [3]Department of Ecology and Evolutionary Biology, University of Kansas, Lawrence, KS 66045, USA. [4]The Institute of Ecology and Evolution, University of Oregon, Eugene, OR 97403, USA. [5]Department of Geography, University of Zurich, Winterthurerstrasse 190, 8057 Zürich, Switzerland. [6]Environmental Studies Program, University of Kansas, Lawrence, KS 66045, USA. ✉e-mail: wanggz@cau.edu.cn; jbever@ku.edu

pathogens in a particular system one selects; and even if influential pathogens are chosen, their effects may depend on interactions with ones that are omitted[15]. Moreover, studies of pathogen dilution classically focus on patterns of pathogen symptoms[8,9,14], while hypotheses of pathogen mediation of biodiversity-productivity relationships depend on pathogen dilution alleviating adverse impacts on plant growth. An alternative and complementary approach to investigating the importance of pathogens focuses on the net impacts on plant fitness from host-specific changes in microbial composition[16,17]. This plant-soil feedback (PSF) framework starts with the observation that pathogen growth rates are host specific, and as a result, pathogens differentially accumulate on particular hosts. Although the effects of multiple pathogens are difficult to evaluate, the net effect of pathogen dynamics on plant-plant interactions can be characterized using a fully reciprocal inoculation experiment[17]. Analysis of this dynamic has revealed that pathogens can contribute to plant species coexistence when intraspecific effects are more strongly negative than interspecific effects[17,18]. That is, pathogen dynamics can contribute to plant species coexistence when the fitness of the plant species declines as a result of pathogen accumulation on their own species (i.e., there is net pairwise negative feedback), as has been found to be common in native plant communities[19,20].

The stabilizing influence of pathogen generated negative feedback might also contribute to statistical complementarity between different plant species (i.e., any changes in the average relative yield in mixture)[21], thereby resulting in overyielding[10]. Specifically, accumulation of host-specific pathogens could reduce productivity in monocultures, while the deleterious impacts of specialized pathogens decrease in diverse communities because of reduced densities of compatible hosts. Manipulations of microbiome composition in mesocosms[22,23] and in the field[24] provide direct evidence of pathogen mediation of overyielding and negative plant-soil feedback models

have predicted patterns of overyielding[25,26]. To date, however, no study has demonstrated the causal connections between host-specific pathogen accumulation and the strength of negative feedback, then connecting the strength of those negative feedbacks with the magnitude of complementarity effects observed in the field. Knowledge of the causal mechanism could inform predictions regarding the context dependence of biodiversity-productivity relationships. For example, as both pathogen-driven feedback and the benefits of pathogen dilution build on host specificity of pathogens, pathogen dilution mediated complementarity should be more common in phylogenetically diverse plant communities, as phylogenetically similar plant species are more likely to share pathogens[27,28]. Moreover, pathogen dilution should increase with plant species richness, so too should pathogen dilution-driven complementarity.

We tested the possibility that feedbacks between plants and pathogens are important drivers of plant diversity generated productivity through an integrated set of field manipulations, greenhouse assays, and feedback modeling (Fig. 1). We planted 18 common prairie plants representing three families in the field experiment (see "Methods"). The design altered plant species richness levels (1, 2, 3, 6), manipulating plant phylogenetic relatedness in mixture (selecting 2, 3, and 6 species from a single family or multiple families). Four months after establishment, we collected soil and root samples, which were sequenced for bacteria, fungi, arbuscular mycorrhizal fungi (AMF), and oomycete (commonly plant pathogens) communities. To determine microbiome feedbacks and pathogen dilution effects, PSF experiments were conducted using the field soil samples as inocula. The next year, we calculated overyielding and complementarity in mixtures and assessed whether the PSF and pathogen dilution predicted biodiversity effects. We then explored the potential of PSFs and pathogen dilution to promote long-term community coexistence and overyielding. Specifically, we parameterized a general feedback model with

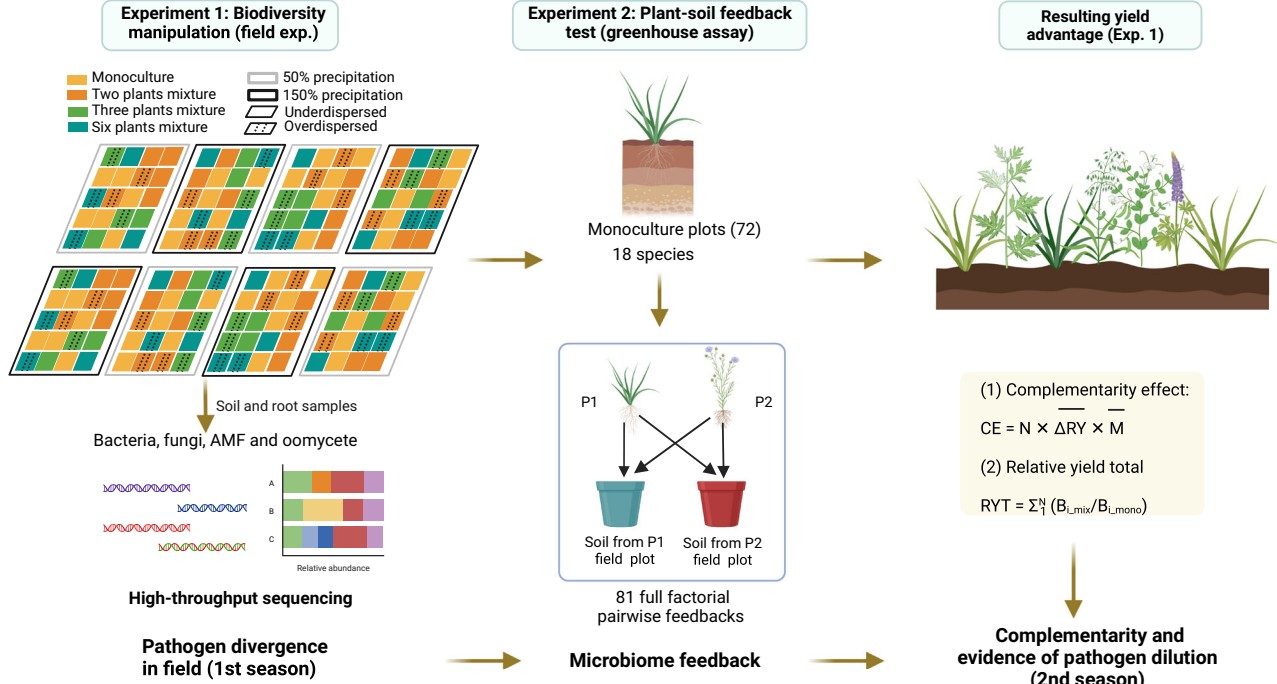

**Fig. 1 | Flow diagram of the experimental design.** A biodiversity manipulation experiment altering plant richness (1, 2, 3, 6) and phylogenetic dissimilarity (underdispersed from the same family, over-dispersed from the multi-families) were established at the University of Kansas Field Station in 2018 (1st season). Both soil and root microbial communities (bacteria, fungi, AMF and oomycete) were sequenced to test the pathogen divergence between different species (Experiment 1). Monoculture soils from the biodiversity manipulation experiment were collected as inocula for greenhouse assays, in which all 18 species were grown in their own or heterospecific soils in a full factorial design to test the soil microbiome feedbacks (Experiment 2). Next, subsequent yield advantages were determined for experiment 1 by calculating the complementarity effect using biomass collected in 2019 (2nd season), to test the underlying mechanism of pathogen dilution effect. Created with BioRender.com.

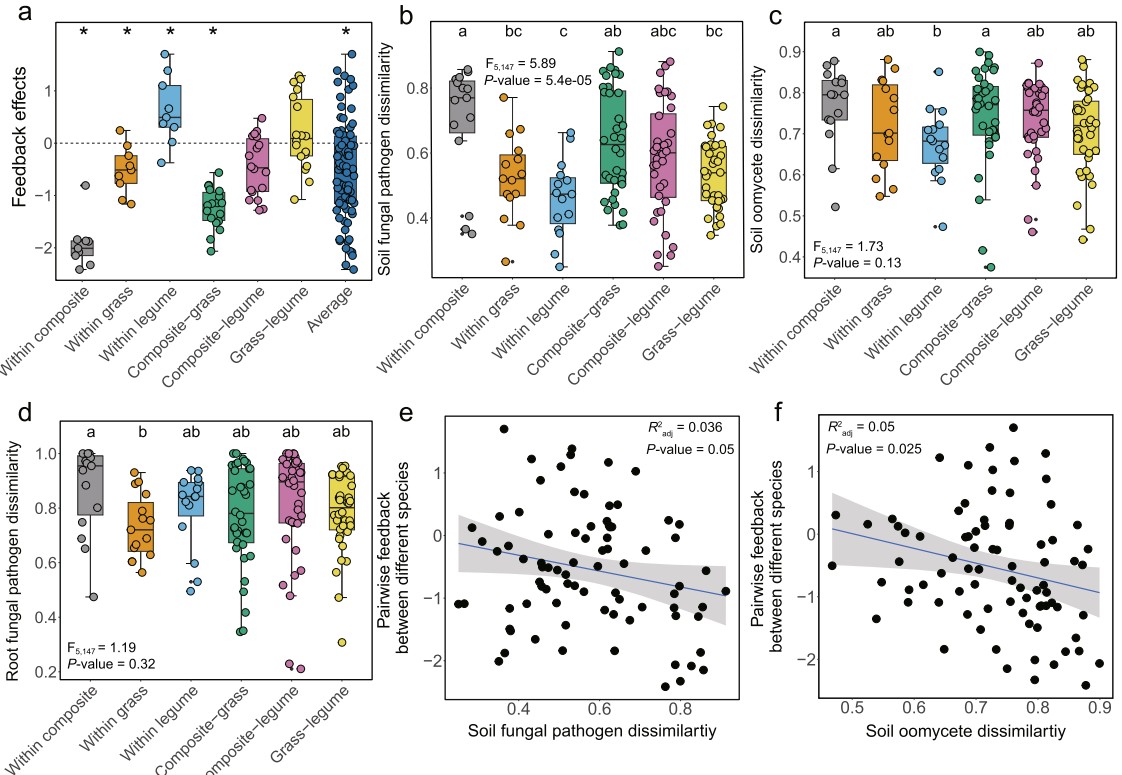

**Fig. 2 | Pairwise feedbacks, soil pathogen dissimilarities and their relationships.** Pairwise feedback effects (**a**), soil fungal pathogen dissimilarity (**b**), soil oomycete dissimilarity (**c**) and root fungal pathogen dissimilarity (**d**) within and between families, respectively. Regressions between soil fungal pathogen (**e**) and oomycete dissimilarities (**f**) and measured pairwise feedbacks. For the number of each group in (**a**), within and between family group $N = 9$ and $N = 18$, respectively; for the number of each group in (**b**–**d**), within and between family group $N = 15$ and $N = 36$, respectively. Asterisks in (**a**) indicate that the PSF significantly differed from zero (95% confidence intervals). Boxplots (**a**–**d**) indicate median (box center line), 25th (box bottom line), 75th (box top line) percentiles, and 5th (the lower whisker) and 95th (the upper whisker) percentiles. Different letters between boxes (**b**–**d**) indicate significant differences between combinations based on two-sided tests ($\alpha = 0.05$, Kruskal–Wallis test, followed by a Dunn's post hoc test). The solid blue lines in (**e**) and (**f**) indicate the fitted relationships and the gray backgrounds indicate the 95% confidence intervals. The statistical test used is *F*-test based on one-sided test, and $p < 0.05$ denotes the overall significance of the regression model. $N = 81$. Source data are provided as a Source Data file.

the empirically derived plant-microbiome interaction strengths (Supplementary Fig. 1)[18]. We hypothesized productivity benefits will be realized in diverse plots and these will be greater where pairwise PSF are more negative, as expected when pathogen dilution drives productivity benefits from diversity.

## Results and discussion
### Pathogens drive negative feedbacks
We observed rapid differentiation of soil pathogens in response to plant community composition 4 months after establishment of the field experiment, and subsequent greenhouse assays confirmed this pathogen divergence generated negative pairwise feedbacks (Fig. 2). On average, pairwise plant-soil feedbacks were negative (Fig. 2a and Supplementary Table 1) (95% confidence interval (CI) −0.48 to −0.24, $p < 0.0001$). While we did not find stronger feedback between families than within families as expected from meta-analyses[19] and phylogenetic structure to pathogen specialization[29], we did observe different feedback patterns between species of different families. PSFs were on average negative between species within the composite (95% CI −0.90 to −1.04, $p < 0.0001$) and grass (95% CI −0.71 to −0.09, $p = 0.011$) families, and between families when composites and grasses (95% CI −1.42 to −0.87, $p < 0.0001$) were grown together (Fig. 2a and Supplementary Fig. 2a). The combination of composites and legumes showed negative PSFs with marginally significant effect (95% CI −0.55 to −0.03, $p = 0.079$) while grasses and legumes combinations showed neutral effect (95% CI −0.20 to −0.26, $p = 0.81$). However, PSFs were on average positive between species within the legume family (Fig. 2a and

Supplementary Fig. 2a) (95% CI 0.23 to 0.89, $p = 0.001$), which is consistent with the limited pathogen specialization within this family (Fig. 2b, c and Supplementary Fig. 2b, c), and specialist nitrogen fixing rhizobia[30].

Soil fungal and oomycete pathogen dissimilarities mirrored patterns observed with negative feedbacks (Fig. 2a–d). Pairs of species within composites (Fig. 2b–d and Supplementary Fig. 2b–d) and grasses (Fig. 2c and Supplementary Fig. 2c), as well as between composites and grasses, and between composites and legumes (Fig. 2b, c and Supplementary Fig. 2b, c) showed greatest average pathogen dissimilarities. Although bacteria and mycorrhizal fungi may contribute to plant-soil feedbacks[31], pathogen dissimilarities were the only components of the microbiome to predict pairwise feedback. Pairwise PSF values were significantly decreased with both pairwise soil fungal ($R^2_{adj} = 0.036$, $p = 0.05$) and oomycete ($R^2_{adj} = 0.05$, $p = 0.025$) pathogen dissimilarities (Fig. 2e, f), but not with other microbial groups (Supplementary Table 2). The best model based on multi-model inference suggested soil fungal pathogen ($p = 0.037$), soil oomycete ($p = 0.028$) and root fungal pathogen ($p = 0.049$) were the strongest predictors of PSF, as they occupied the highest weight in the model (Supplementary Table 3). These findings are consistent with results of a meta-analysis showing that PSF effects are generally stronger and more negative when pathogens are included in the soil community[19]. While we cannot identify the contribution of individual pathogens to PSFs, we observed the most abundant fungal pathogens differed for most plant species (Supplementary Fig. 3), which suggests that candidate pathogens may drive negative feedbacks. By showing that

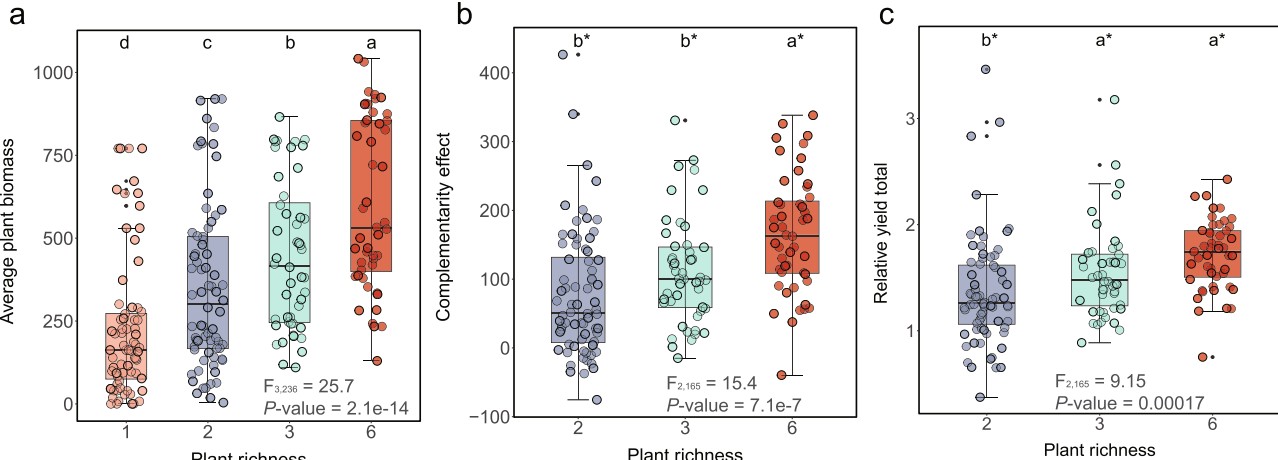

**Fig. 3 | Plant productivity and biodiversity effects by planted richness.** Average plant biomass (g m$^{-2}$) (**a**), complementarity effect (**b**) and relative yield total (**c**) by plant richness in the 2nd season of the biodiversity manipulation experiment (2019). Monocultures and two species mixtures each contain 72 plots and three and six species mixtures 48 plots each. Boxplots indicate median (box center line), 25th (box bottom line), 75th (box top line) percentiles, and 5th (the lower whisker) and 95th (the upper whisker) percentiles. Different letters between boxes (**b**, **c**) indicate significant differences between combinations based on two-sided tests ($\alpha = 0.05$, Kruskal–Wallis test, followed by a Dunn's post hoc test). Asterisks in (**b**) and (**c**) indicate that complementarity and relative yield totals in richness levels 2–6 were significantly greater than 0 and 1, respectively. Source data are provided as a Source Data file.

pathogen-driven negative feedbacks can be generated over one growing season in the field, our results add to the growing evidence that accumulation of host-specific pathogens contribute to plant species coexistence[19,20,32].

### Negative feedbacks predict plant diversity benefits

We found above-ground productivity increased with planted species richness ($F_{3,236} = 25.7$, $p < 0.0001$) in the second season of the field experiment (Fig. 3a). In plant mixtures, both complementarity[21] ($F_{2,165} = 15.4$, $p < 0.0001$) and relative yield totals[33] ($F_{2,165} = 9.15$, $p = 0.00017$) increased with richness (Fig. 3b, c and Supplementary Table 4), and the relative yield totals were strongly driven by positive complementary effects ($R^2_{adj} = 0.56$, $p < 0.0001$) rather than by species-specific selection effects ($R^2_{adj} = 0.0034$, $p = 0.45$) (Supplementary Fig. 4). Despite the expectation that legumes should increase resource partitioning given their ability to access nitrogen, an often limiting nutrient in terrestrial systems[2,34], we did not observe consistently stronger complementarity in mixtures including legumes (multiple-families) compared to those including species of the same family (Supplementary Table 5). While stronger complementarity in mixtures including species of different families was also expected from phylogenetic structure of pathogen specialization[28], we found similar levels of pathogen dissimilarities and feedback strengths within and between families (Fig. 2 and Supplementary Table 6), qualitatively matching the complementarity results.

Complementarity increased with the strength of pathogen-driven plant-soil feedbacks (Fig. 4). We quantified plant-soil feedbacks by weighted average pairwise feedback (Eq. (4) in the "Methods") extrapolated from pathogen dispersion measures (Eq. (3)). With more negative pairwise PSF, statistical complementarity ($R^2_{adj} = 0.11$, $p < 0.0001$) and productivity benefits ($R^2_{adj} = 0.11$, $p < 0.0001$) in diverse plots significantly increased in the second growing season (Fig. 4a, c). These relationships held in both single- and multi-family mixtures, as well as in plots with nitrogen-fixing legumes and plots without (Supplementary Fig. 5, Supplementary Table 7 and Supplementary Discussion), which suggests pathogen dilution is generalizable and consistent mechanism generating productivity responses to diversity. The change in relative abundance of the most abundant pathogen decreased with plant richness providing further evidence pathogen dilution (Supplementary Figs. 6–8 and Supplementary

Discussion). Together, this causal chain of specialist pathogens generating negative feedback which predicts yield increases in mixture, demonstrates that pathogen dilution generates overyielding. Accounting for the pathogen dilution expected from increased species richness (Eq. (5)) revealed an even stronger positive relationship with observed complementarity ($R^2_{adj} = 0.12$, $p < 0.0001$) and relative yield total ($R^2_{adj} = 0.11$, $p < 0.0001$) (Fig. 4b, d).

### Feedback modeling corroborated the pathogen dilution mechanism

Simulations of long-term community dynamics demonstrate pathogen dilution can generate linear relationships between complementarity and plant species richness (Fig. 5a), as observed in long-term biodiversity manipulations[35,36]. Analysis of a general feedback model[18] parameterized using observed pairwise pathogen dissimilarities identifies pathogen-driven feedbacks can both stabilize diverse communities and generate positive productivity responses to diversity that persist over time (Fig. 5, Supplementary Figs. 9–11 and Supplementary Discussion). The simulations provide additional substantiating evidence that the observed dependence of complementarity in the field on average pairwise feedbacks from greenhouse assays (Fig. 4a, c) are signatures of pathogen dilution, an important mechanism that can theoretically replicate long-term community dynamics; thus linking field observations with coexistence models (Fig. 5b, c). Moreover, the theoretical results indicate this relationship has two components: (1) within each species richness level, a positive relationship with the strength of negative feedback, and (2) across species richness levels, a stair step progression toward higher average complementarity and more strongly negative average feedbacks (Fig. 5b). Together, these components generate an asymmetric tail associated correlation[37]. We find that our empirical data (Fig. 4a) share these features (Supplementary Figs. 9–11), providing additional evidence that pathogen dilution drives complementarity in our field experiment.

Together, our empirical and theoretical results provide strong evidence that specialist pathogen dilution causally generates complementarity and overyielding. While the ecological importance of biodiversity diluting pathogens has been controversial given devastating examples of pathogen spillover between animal populations[38,39], we find strong support that the diversity of native plants reduce deleterious impacts of specialist pathogens on plant productivity. We

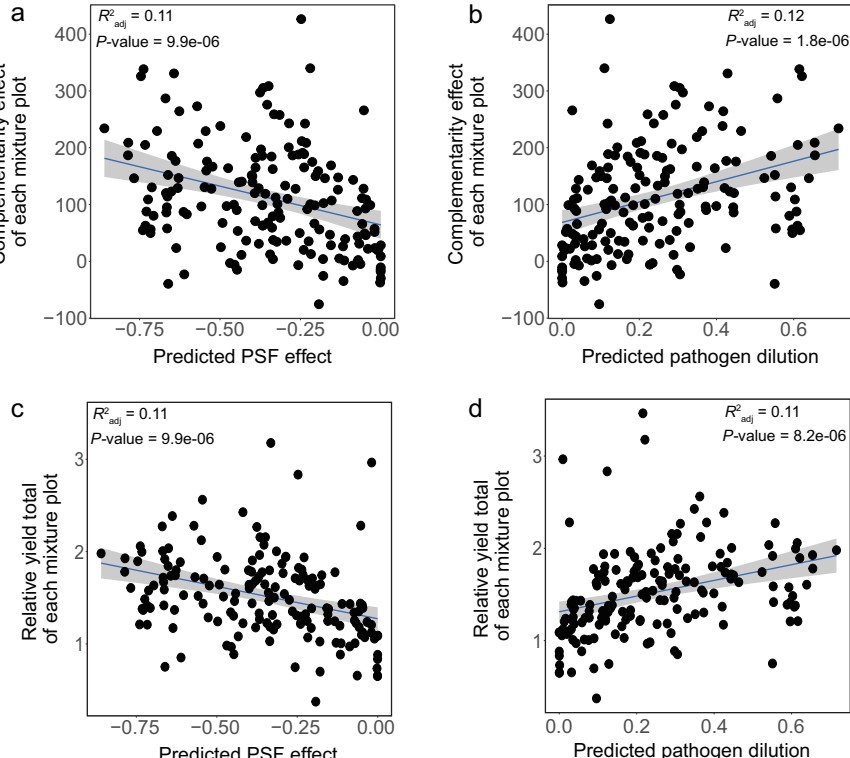

**Fig. 4 | The relationship between plant productivity and plant-soil feedback and pathogen dilution.** Regressions between complementarity and predicted PSF effect (**a**) and pathogen dilution effect (**b**). Regression between relative yield total and predicted PSF effect (**c**) and pathogen dilution effect (**d**). The solid blue lines indicate the fitted relationships and the gray background indicates the 95% confidence interval. $N = 168$. Source data are provided as a Source Data file.

observed rapid differentiation of soil pathogens in response to changes in plant diversity and community composition 4 months after plant establishment in the field and this pathogen divergence generated negative pairwise microbiome feedbacks (Fig. 2). With higher negative pairwise PSF, we found an increase in complementarity and productivity with diversity the subsequent growing season (Figs. 3 and 4), which is consistent with pathogen dilution contributing to positive productivity responses to plant diversity. Parameterizing a general theoretical model with our empirical observations corroborates that these relationships are causal, generalizable, and capable of generating linear relationships between plant diversity and productivity over time (Fig. 5)[35,36]. Our results provide the strongest evidence to date that dilution of host-specific pathogens contribute to plant diversity driven yield advantages, consistent with previous work manipulating pathogen abundance in field[24,40] and mesocosm experiments[22,23], as well as studies that connect negative feedback results to positive diversity-productivity relationships[23,41]. While more work is necessary to evaluate the generality of pathogen dilution across systems, our findings indicate pathogen dilution can independently generate patterns consistent with observations of productivity increases with plant diversity (i.e., biodiversity-ecosystem functioning, BEF, relationships)[2-4].

When and where pathogen dilution generates BEF relationships will likely depend on ecological context. This mechanism may be particularly important for systems containing coevolved plants and pathogens; for example, negative feedbacks are commonly observed between native plants, but not between non-native plants[19]. Consistent with this hypothesis, biodiversity manipulation experiments with non-native plant species have shown low levels of complementarity compared with native plant species[42,43]. We included native prairie soil microbiome at the initiation of our experiment manipulating native prairie plants, which may have made our detection of pathogen dilution more likely. This could also explain why we observed significant

complementarity in the first 2 years of the biodiversity manipulation, while complementarity took multiple years to realize in other BEF experiments[34,35]. Many biodiversity manipulations were initiated with soil microbiomes degraded by anthropogenic disturbance, including conventional agriculture. As a result, it may have taken several years for specialist pathogens to reestablish at those sites. In addition, global change drivers such as warming[44], high precipitation[45] and nutrient enrichment[46] can have positive effects on pathogens, and therefore may strengthen BEF relationships. While we show that pathogen dilution can rapidly generate productivity gains from diverse plant communities, more work is necessary to link these productivity benefits of pathogen dilution to other positive ecosystem responses that may occur with plant diversity, such as the increased carbon sequestration[47,48]. Moreover, our theoretical results suggest that pathogen dilution may promote community coexistence, meaning that productivity increases can also be maintained over longer periods of time.

While resource partitioning has historically been thought to be the principal force structuring plant communities, our work suggests that dynamics between host-specific pathogens and plants can contribute to species coexistence and BEF relationships. Further work is necessary to explore the relative importance of these two forces in structuring plant communities and functioning. Moreover, more work on the importance and context dependence of biotic PSFs and resources partitioning would improve understanding of the interactive forces of biodiversity loss and anthropogenic change.

## Methods
### Experiment 1: field experiment and measurement
The field experiment was established in May 2018 at the University of Kansas Field Station (39°03'09" N, 95°11'30" W), located in eastern Kansas, USA. This land historically was tallgrass prairie, but was tilled for row crops or pasture from 1870 to 1970, after which it was left

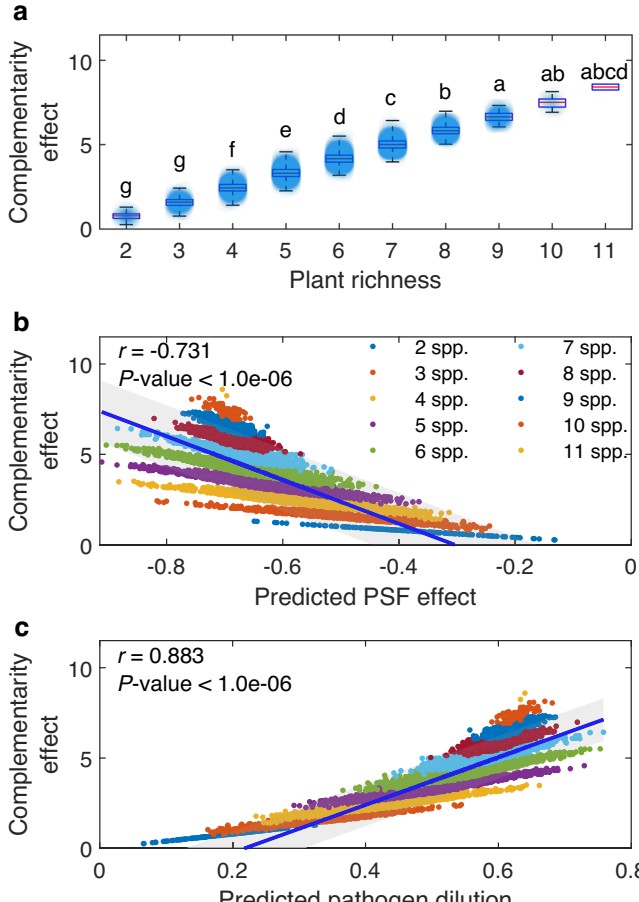

**Fig. 5 | Analytical solutions of a general feedback model parameterized with the empirically derived interaction strengths between the plant species.** These model results reveal the expected relationships between complementarity, species richness, predicted plant-soil feedback (PSF) effects and predicted pathogen dilution when plant community dynamics are driven by host-specific soil pathogens. These include generating a positive relationship between complementarity and plant richness (**a**), a negative relationship with predicted plant-soil feedback (**b**) and a positive relationship with expected pathogen dilution (**c**). In (**a**), lines indicate group medians, box edges indicate 25th and 75th percentiles, whiskers indicate the minimum and maximum values within the observation group. In the latter relationships, this relationship shows an asymmetric tail association correlation that is consistent with the empirical data (see Supplementary Information for details). Different letters between boxes indicate significant differences between plant richness levels ($\alpha = 0.05$, Kruskal–Wallis test, followed by a Dunn's post hoc test). In (**b**) and (**c**), the solid blue lines indicate the fitted relationships for mean complementarity effects, with the gray background indicating the 95% confidence interval. $N = 17,850$. Source data are provided as a Source Data file.

fallow and was inhabited by non-native pasture grasses, with native prairie grass establishment occurring in the last 10 years. Prior to planting, we tilled the resident soil and tilled in native prairie soil as inocula for the soil microbiome (see Supplementary Methods). A total of 240 plots (1.5 m × 1.5 m) were established, with 18 native prairie plant species chosen from three families (six species each from *Poaceae*, *Fabaceae*, and *Asteraceae*, see Supplementary Table 8). We manipulated species richness (1, 2, 3, and 6), plant community composition (phylogenetically under or over dispersed), and precipitation (50 or 150% ambient rainfall) (Fig. 1). In plant mixtures, under-dispersed treatments contained plants from a single family while over-dispersed treatments contained plant species from two or three families. The full-factorial design comprised 72 monocultures, 72-two species, 48-three species, and 48-six species plots, with each of the 18

plant species being equally represented in all treatment combinations. We established six subblocks containing 40 paired plots distributed across replicate rainfall exclusion shelters containing 20 plots with the same plant combinations (6-monocultures, 3-each of phylogenetically over- and under-dispersed two species mixtures, 2-each of phylogenetically over- and under-dispersed three species mixtures, 2-each of phylogenetically over- and under-dispersed six species mixtures). Beginning in spring 2019, one paired precipitation exclusion shelter received 150% ambient growing season rainfall, while other paired shelter received 50% ambient growing season rainfall. To help with plant establishment and since construction of the rainfall exclusion shelters was not completed until August 2018 and both precipitation treatments received ambient rainfall to August 2018. As a result, we did not consider precipitation effects on microbial community composition in the current study.

Soils were collected from each plot in September 2018, 4 months after planting, and then paired plots of matched plant composition were pooled across paired rainfall exclusion shelters. A total of six 20 cm soil cores with 1.9 cm diameter (~340 ml) were taken from each plot, sealed in polyethylene bags and kept on ice and then transferred to the lab for storage. The coring devices were sanitized between different treatments to minimize contamination. The soil cores were taken close to planted species to be more representative of plant-affected soil microbes and to ensure we were able to collect root samples. The soil samples were separated into two subsamples. One subsample (~50 ml) was passed through a 4-mm mesh to separate roots, and then both soil and sterile water washed root samples were stored in a freezer at −20 °C for DNA extraction and sequencing. The remaining soil was stored at 4 °C until it was used for the subsequent plant-soil feedback experiment.

In July 2019, at peak biomass, plant aboveground biomass was harvested and weighed from 0.1 m² strips and plant cover surveys were conducted for the total plot. Biomass was sorted to species within 1 day, dried at 70 °C for at least 3 days, and weighed. Strong correlations between plant cover and biomass (Pearson's $R = 0.76$; $t_{[646, 0.05]} = 29.6$; $p < 0.001$) allowed us to develop species-specific regressions to convert cover to biomass and present community-level biomass yields at a plot level.

### Experiment 2: plant-soil feedback effect test and calculation

Field monoculture soils collected in Experiment 1 were used as inocula to test both conspecific and heterospecific plant-soil feedback (PSF) effects (Fig. 1 and Supplementary Fig. 12). The background soil (20 cm depth) was collected from the Field Station with a pH 5.93, 0.17% total nitrogen, 6.7 mg kg⁻¹ Mehlich phosphorus, and 3.83% organic matter. The soil was passed through a 4-mm sieve to remove stones and coarse roots and then mixed up 1:1 with river sand. The mixture of soil and sand as background soil was steam sterilized twice for 4 h with a 1-day rest period between sterilization. Each deep-pot (diameter 6.4 cm, height 25.4 cm) was filled with 500 ml soil into three layers: 225 ml sterile soil at the bottom, 50 ml (10%) inoculum in the middle, and another 225 ml sterile soil on the top.

We used all 18 species and grew them with pots inoculated with soils from conspecific monocultures, or monocultures from other species from the same family, or from a different family. Each of the 18 species were grown with inocula from three plant species monoculture plots from each family (Supplementary Fig. 12), which resulted in 81 full factorial pairwise feedback tests. For each of 18 species, one individual of each species was planted, with their own monoculture soils (nine replicates) and nine soil treatments for other species (three replicates of three heterospecific species from each family), and sterile soil was used for controls, including three replicates for each species. In total, there were 702 pots and all pots were arranged in a randomized block design. Greater replication of conspecific plantings allows greater confidence in measurement of this effect which is used for

multiple pairwise feedback calculations involving individual species[49]. The experiment was carried out in a glass greenhouse at University of Kansas, Lawrence, USA. The natural light intensity was ~800–1200 $\mu$mol m$^{-2}$ s$^{-1}$ with a night-day temperature range of 20–30 °C. All pots were irrigated with daily with drip irrigation system and the pots were measured every 3 days to maintain the soil moisture content at ~20% (w/w). Plants were harvested after 2 months of growth, separating and weighing shoot and root dry biomass.

To measure PSF effects between species, we calculated the log response ratio of pairwise PSF[19]:

$$PSF = \ln\left(\frac{\alpha_A}{\beta_A} \Big/ \frac{\alpha_B}{\beta_B}\right) = \ln\left(\frac{\alpha_A}{\beta_A}\right) \ln\left(\frac{\alpha_B}{\beta_B}\right) = \ln(\alpha_A) + \ln(\beta_B) - \ln(\beta_A) - \ln(\alpha_B)$$

(1)

where $\alpha_A$ is species $A$'s average performance grown in species $A$'s soil community ($\alpha$), $\beta_B$ is species $B$'s average performance grown in species $B$'s soil community ($\beta$), and $\alpha_B$ and $\beta_A$ are the average performances of species $B$ and $A$ grown in soil $A$ and $B$, respectively. This formula was modified from the classic pairwise PSF formulation ($I_s = \alpha_A + \beta_B - \alpha_B - \beta_A$)[18], which compares the relative difference of the two plant species' performances in their conspecific soils versus heterospecific soils. In both formulations of pairwise PSF, negative values are required for plant coexistence while positive values prevent coexistence.

## DNA extraction and sequencing processing

Microbial DNA of both soil and root samples were extracted from all the field treatments. DNA were extracted from 0.25 g fresh soil following the manufacturer's instructions (DNeasy PowerSoil Kit, Qiagen, Hilden, Germany) as well as 0.25 g of roots separated from soil. Bacterial, fungal, oomycete, and AM fungal communities were sequenced from both soil and root DNA. The bacterial primers (forward 515F, 5′-GTGYCAGCMGCCGCGGTAA-3′; reverse 806R 5′-GGAC-TACNVGGGTWTCTAAT-3′) targeting the V4 region of 16S small subunit (SSU) of ribosomal RNA[50,51], fungal primers (forward fITS7, 5′-GTGAGTCATCGAATCTTTG-3′; reverse ITS4, 5′-TCCTCCGCTTATTGA-TATGC-3′) targeting the internal transcribed spacer (ITS) region[52], mycorrhizal fungal primers (forward fLROR, 5′-ACCCGCTGAACT-TAAGC-3′; reverse FLR2, 5′-TCGTTTAAAGCCATTACGTC-3′) designed for the large subunit (LSU) region[53] and oomycete primers (forward ITS300, 5′-AGTATGYYTGTATCAGTGTC-3′) and reverse ITS4)[54] targeting the ITS region were selected for polymerase chain reaction (PCR). For bacteria, fungi and AMF, PCR reactions were conducted in a final volume of 25 $\mu$l with 1 $\mu$l template DNA, 10.5 $\mu$l ddH$_2$O, 0.5 $\mu$l of forward and reverse primer and 12.5 $\mu$l of Master Mix Phusion (Thermo Fisher Scientific, Waltham, MA, USA). For oomycete, the reactants in each sample were 0.5 $\mu$l of both primers, 1.0 $\mu$l of template DNA, 5.0 $\mu$l HOT FIREpol (Solis Biodyne, Tartu, Estonia), and 18 $\mu$l of ddH$_2$O. The bacterial PCR program was as follows: 94 °C for 5 min; 35 × (94 °C for 30 s; 57 °C for 30 s and 72 °C for 30 s); ending with 72 °C for 7 min. The mycorrhizal fungal PCR program was: 94 °C for 5 min, 35 × (94 °C for 30 s, 48 °C for 30 s, and 72 °C for 45 s), ending with 72 °C for 10 min. The oomycete PCR program was: 5 min at 95 °C, 35 × (30 s at 95 °C, 30 s at 55 C°, 60 s at 72 °C), ending with 72 °C for 10 min. Four microliters of PCR product were checked on 1.5% (w/v) agarose gel to estimate the quality of PCR products. PCR products were barcoded using Nextera XT Index Kit v2 (Illumina, San Diego, CA, USA) for indexing and purified using AMPure XP beads (Beckman Coulter, Brea, CA, USA) before sequencing. For bacteria, fungi and AMF, barcode PCR cycle began at 98 °C for 30 s; 10 × (98 °C for 10 s; 55 °C for 30 s and 72 °C for 30 s), ending with 72 °C for 5 min. For oomycete, the PCR was run under similar conditions as the initial PCR, except 5 $\mu$l of the primary PCR amplicon was used instead of the original DNA template. PCR products concentration was measured by Invitrogen Qubit 3.0 Fluorometer

(Thermo Fisher Scientific, Waltham, MA, USA). Samples were pooled in equimolar concentration to a single library for each target group. Sequencing was performed by Illumina MiSeq v3 PE300 Next-Gen Sequencer in Genome Sequencing Core of University of Kansas.

After sequencing, the primary analysis of raw FASTQ data was processed with the QIIME2 pipeline[55]. After sequences were demultiplexed and primers removed, they were quality filtered, trimmed, denoised, and merged using DADA2[56]. Taxonomy was assigned to all ribosomal sequence variants in QIIME2 using a feature classifier trained with the SILVA 99% OTU database for bacteria[57] and the UNITE 99% database for fungi (Version 18.11.2018)[58]. The mycorrhizal fungal database was further examined to discard sequences falling outside the AM fungal clade by creating a phylogenetic tree using a self-constructed reference database and pipeline[53]. These sequences were aligned using MAFFT[59] and a phylogenetic tree constructed using RAXML v.8[60] with 1000 bootstrap replicates and the evolutionary model GTRGAMMA in QIIME2 (2019.10). Outgroups were *Mortierella elongata* (MH047197, Mucoromycota), *Exophiala spinifera* (MH876260; Basidiomycota) and *Rhodotorula hordea* (AY631901; Ascomycota), and LSU sequences of a plant, *Citrus limon* (X05910, Rutaceae), and an animal, *Rutilus rutilus* (EF417167, Cyprinidae) were also included. Functional groups within the fungal community were categorized based on the FungalTraits database[61]. The amplicon sequence variants (ASVs) categorized as "plant_pathogen" in the "primary_lifestyle" or "secondary_lifestyle" were used as putative pathogens, and "soil_saprotroph", "litter_saprotroph", "wood_saprotroph" and "unspecified_saprotroph" in the "primary_lifestyle" were used as saprobes for subsequent analyses. For soil, 1904 out of 7272 ASVs were matched to a guild, and 212 of those were putative pathogens. For roots, 139 of 2825 ASVs were matched to a guild, and 133 of those were putative pathogens. For oomycetes, we checked the identity of resulting OTUs either against a database containing all NCBI oomycote ITS2 sequence results using the Basic Local Alignment Search Tool, BLAST v. 2.6.0[62], using default parameters, or through placing OTUs in the oomycete clade, as the oomycota are thought to have arisen from a common ancestor forming a conserved clade[63]. The terrestrial oomycetes are primarily parasites of vascular plants[64,65] and generally function as pathogens. All samples were normalized to a sequencing depth of lowest total reads per sample (soil fungi-3880, root fungi-3943, soil bacteria-661, root bacteria-658, soil AMF-860, root AMF-358, soil oomycete-2004, root oomycete-200) prior to downstream analyses in R (version 3.5.1) prior to downstream analyses by using the rarefy function in the *vegan* package[66] in R (version 3.5.1).

## Statistical analysis

We assessed the ecological importance of soil microbiome composition in generating PSFs by comparing microbial dissimilarities with pairwise PSF effects between different species. In Experiment 1 using field data, we first analyzed microbial composition differences by calculating pairwise Bray–Curtis dissimilarities between pairwise species monocultures using the *vegan* package[66] in R. For the analysis of Experiment 2, plant biomass was log-transformed to improve homoscedasticity and explained using a linear model with plant species, inocula species, and their interactions as fixed effects. Seedling height at planting by plant species was included as a covariate to remove the effect of initial size differences. Marginal means and standard errors of each plant species by inocula species pair were estimated from the model and were used to calculate pairwise feedback effects between species using Eq. (1). We compared the PSF effects within and between families, namely the pairwise PSF effect between the host plant and the other species from the same family or other families, respectively. To test whether overall PSF was significantly different from zero, the *rma.mv* function as implemented in the *metafor* package[67] was used. We fit a random effects model using PSF (Eq. (1)) as the response variable with pairwise combination as a random effect and the

individual estimate variance (Var$_{PSF}$) as the variance[19]. The Var$_{PSF}$ was calculated as:

$$Var_{PSF} = \frac{Var_{\alpha_A}}{N_{\alpha_A} \times (\alpha_A)^2} + \frac{Var_{\beta_B}}{N_{\beta_B} \times (\beta_B)^2} + \frac{Var_{\alpha_B}}{N_{\alpha_B} \times (\alpha_B)^2} + \frac{Var_{\alpha_B}}{N_{\alpha_B} \times (\alpha_B)^2} \qquad (2)$$

where Var and $N$ represent variance and sample size, respectively. As conspecific means were used for multiple pairwise feedback estimates, the sample size of conspecifics, $N_{\alpha_A}$ and $N_{\beta_B}$, were adjusted in the Var$_{PSF}$ calculation for the nine pairwise feedbacks in which they were used from the 9 conspecific replicates in the experiment to 1 (=9 replicates/9 pairwise feedback estimates). The PSF effects were considered significantly different from zero if the 95% confidence interval did not contain zero.

The regressions between different microbial community dissimilarities and measured pairwise PSFs were calculated and the regression coefficients and the functions' significance were obtained with the *stat_poly_eq* and *stat_fit_glance* functions in the *ggpmisc* package[68]. The regressions between soil fungal pathogen and oomycete dissimilarities and measured pairwise PSFs were significant (Supplementary Table 2), indicating the importance of fungal pathogens in driving negative PSFs. We then considered two different approaches (linear model and random forest) to predict pairwise feedback using 12 potential microbial predictors (rhizobia, non-rhizobia bacteria, AMF, fungal pathogen, fungal saprobe and oomycetes for both soil and root). However, for soil rhizobia, only 28 ASVs were identified and there were many missing values when calculating the dissimilarity and could not be used in later modeling running. Therefore, we removed the soil rhizobia when running the models, keeping the other 11 microbial predictors. Linear and random forest models[69] were compared using the *lm* and *randomForest* base R functions, and we assessed the significance of fitted random forest models using the *A3* package[70]. We found that the linear model explained a larger proportion of variance ($R_{adj}^2 = 0.137$) than the random forest model ($R_{adj}^2 = 0.106$). Then, we performed a model selection process for feedback effects, using the *glmulti* package[71] in R to generate a suite of candidate models and sample size corrected Akaike Information Criterion (AIC$_c$)[72] to distinguish between candidate models. As ΔAIC$_c$ between models were statistically indistinguishable (≤2), we opted to perform model averaging to represent the importance of parameters across all candidate models[73]. Parameter weights were estimated as the sum of Akaike weights across all the models[74] and are considered a measure of the overall support for each predictor. We set a lower limit of 0.7 to differentiate between the important and unimportant predictors. We found that soil fungal pathogen, soil oomycete and root pathogen dissimilarities were the three most important predictors and their effects on predicting the PSF were significant (Supplementary Table 3). We then derived a best linear model based on these three predictors to characterize the pairwise PSF effects:

Predicted PSF$_{ij}$ between species $i$ and $j$ = −1.52*soil fungal pathogen dissimilarity
−2.27*soil oomycete dissimilarity
−1.27*root fungal pathogen + 1.589
(3)

where the coefficients of both soil and root fungal pathogen and oomycete dissimilarities were the estimates derived from the predicted model (Supplementary Table 3). To determine the intercept, we calculated a weighted average of all multiple regression model intercepts.

To estimate PSF effects for plant mixtures in the field, the predicted PSF was calculated as the sum for a plot of PSF$_{ij}$ weighted by pairwise plant densities:

$$\text{Predicted PSF effect of plots} = \sum pi\, pj\,(PSF_{ij}) \qquad (4)$$

where $p_i$ and $p_j$ represent the realized density (proportion of the total biomass) observed in the field, and PSF$_{ij}$ represents the pairwise feedback between those two species obtained from Eq. (3).

A more negative PSF indicates greater pathogen suppression in monocultures and therefore greater potential pathogen dilution−i.e., the degree of pathogen dilution should be predicted by pathogen-generated PSF. However, in plant mixtures, pathogen dilution may also depend on plant species richness. For example, the expected chance of the closest neighbor being conspecific is 1 out of 2 in two species mixtures, but only 1 out of 6 in six species mixtures. That is the expected dilution due to heterospecific neighbors is $(N − 1)/N$ (=1 − 1/$N$), where $N$ is the number of species in the plot. We therefore estimate expected release from pathogens for a given plot as the product of average PSF and the expected dilution, times −1 to represent release from the negative feedback:

$$\text{Predicted pathogen dilution} = \text{Predicted PSF} \times -1 \times (1 - 1/N) \qquad (5)$$

We postulate negative PSF and pathogen dilution effects are an underlying mechanism generating complementarity and explaining subsequent yield advantages. We used the second growing season's aboveground biomass (2019) to calculate complementarity (CE) and selection effects (SE)[21]. Complementarity effects are highest where most or many species yields are higher than expected based on yields in monoculture. Selection effects are high when one or few species generate the productivity gains generated from diversity. We calculated these components as follows:

$$CE = N \times \overline{\triangle RY} \times \bar{M} \qquad (6)$$

$$SE = N \times cov(\triangle RY, M) \qquad (7)$$

where $N$ is the plant species richness, ΔRY is deviation from expected relative yield of species in the mixture, $M$ is a species' average monoculture biomass, $\overline{\triangle RY}$ is the mean change in relative yield for each species in mixture, and $\bar{M}$ is the mean monoculture biomass for each species. We also calculate the relative yield total (RYT)[33], which is a measure of the relative productivity of plant mixtures compared to monocultures, and could be used to indicate overyielding, calculated as:

$$RYT = \Sigma_1^N \frac{B_{i\_mix}}{B_{i\_mono}} \qquad (8)$$

where $B_{i\_mix}$ represents observed biomass production of species $i$ in mixture, while $B_{i\_mono}$ represents biomass production of species $i$ in monoculture. RYT > 1 indicates overyielding, whereas RYT < 1 indicates underyielding.

The effects of phylogenetic dispersion (within family, between family), plant richness (2, 3, 6) and their synergistic effects on plant biomass, complementarity effect and relative yield total of mixture plots were tested using the analysis of variance (ANOVA). We used *t*-tests to determine whether complementarity effect and relative yield totals in richness levels 2–6 were significantly greater than 0 and 1, respectively. Relationships between independent variables, predicted PSF effect and predicted pathogen dilution, and dependent variables, complementarity effect and RYT of plant mixtures were analyzed using simple linear regression analysis.

## Theoretical analysis

We utilized empirically derived plant host-microbiome interactions to parameterize a general mathematical model of host-environment feedback (Supplementary Fig. 1). Previous studies have shown how this model can be applied specifically to the study of plant-soil feedbacks, which mediate plant community coexistence[18]. In summary, plant abundances are expressed as proportions or frequencies, with plant-soil microbiome effects driving frequency-dependent feedbacks controlling plant community dynamics. Changes in plant frequencies are described by:

$$\frac{dP_i}{dt} = P_i\left(w_i - \sum_{j=1}^{N} w_j P_j\right) \tag{9}$$

In which:

$$w_i = \sum_{j=1}^{N} \sigma_{ij} P_j \tag{10}$$

where $P_i$ is the frequency of plant species $i$, $w_i$ is its fitness within the current environment, the state of which is determined by the frequencies of all $N$ species within the community. The parameters $\sigma_{ij}$ determine the fitness of plant species $i$ in an environment dominated by plant species $j$. We parameterized the coefficients $\sigma_{ij}$ using the empirically quantified dissimilarities in fungal pathogen, root fungal pathogen and oomycete compositions of plant hosts and Eq. (3). Note that conspecific effects could be calculated assuming that pairwise feedback between two plant species with completely dissimilar soil microbiomes is the sum of the two conspecific effects, which can be calculated from Eq. (3) with dissimilarity = 1. Furthermore, we assumed that the strength of conspecific effects were equal among all plant species. This approach yielded a constant fitness, $w_{mono}$, for all species growing in their own soil environment. Quantification of all the $\sigma_{ij}$ parameters then enabled the construction of an interaction matrix:

$$\mathbf{A} = \begin{bmatrix} \sigma_{11} & \sigma_{12} & \ldots & \sigma_{1N} \\ \sigma_{21} & \sigma_{22} & \ldots & \ldots \\ \ldots & \ldots & \ldots & \ldots \\ \sigma_{N1} & \sigma_{N2} & \ldots & \sigma_{NN} \end{bmatrix} \tag{11}$$

Which determines the feasibility and stability of the community coexistence equilibrium state. Specifically, the feasibility condition can be written as:

$$0 < P_i = \frac{\det \mathbf{A}_i}{\sum_{j=1}^{N} \det \mathbf{A}_j} < 1 \quad \text{for } i = 1, 2, \ldots, N \tag{12}$$

If the feasibility condition is fulfilled, it follows from Eq. (9) that at the community coexistence equilibrium point:

$$\hat{w}_i - \sum_{j=1}^{N} \hat{w}_j \hat{P}_j = 0 \tag{13}$$

where the hats indicate an equilibrium state. From Eq. (13), it can be inferred that for all species[19]:

$$\hat{w}_i = \sum_{j=1}^{N} \hat{w}_j \hat{P}_j = \hat{w} \tag{14}$$

In which $\hat{w}$ is the fitness of all species at the coexistence equilibrium point. Hence, a measure for complementarity is then provided, as the difference between this equilibrium fitness $\hat{w}$ and the fitness of each plant species in monoculture, $w_{mono}$. Complementarity was calculated according to ref. 22 (see Eq. (6)). Within our specific model formulation, this complementarity effect in simulated communities was calculated using relative fitness as a proxy for biomass:

$$CE = N \times \overline{\triangle RY} \times \bar{M} = N \times (\hat{w} - w_{mono}) \tag{15}$$

where we used that $\overline{\triangle RY} = \frac{\hat{w} - w_{mono}}{w_{mono}}$ and $\bar{M} = w_{mono}$. We assessed the feasibility, community-level feedback, and local stability of all 262,125 multispecies communities that could be assembled from the pool of 18 species considered in Experiments 1 and 2 described above. Local stability was indicated by the dominant eigenvalue of the Jacobian matrix evaluated at the coexistence equilibrium point[19]. For all feasible and locally stable communities that exhibited negative community-level feedback (ranging between 2 and 11 species), we computed the predicted PSF effect (Eq. (4)), pathogen dilution (Eq. (5)), and the strength of the complementarity effect (Eq. (15)), to test the extent to which the former two could predict the latter. We note that our selection of communities assesses the potential of plant-soil feedback to enable community coexistence conservatively[17,18]. Alternative approaches suggest a larger potential of plant-soil feedback to generate coexistence. Importantly, the inferred theoretical relationships between predicted PSF effects, pathogen dilution, and complementarity were robust under the different specific community selection approaches taken (Supplementary Discussion).

## Reporting summary

Further information on research design is available in the Nature Portfolio Reporting Summary linked to this article.

## Data availability

All raw data required to reproduce the results are provided in the Figshare[75] (https://doi.org/10.6084/m9.figshare.23804619). Sequences were submitted to the NCBI Sequence Read Archive (SRA) under the accession number PRJNA863284. Source data are provided with this paper.

## Code availability

All codes used in this study are available on GitHub (https://github.com/wlzwgz/Dimension-PSF.git).

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

## Acknowledgements

We acknowledge the KU Field Station for resources and support for this project. We thank K. Mecke for setting up and managing the experiment; the staff at KU Field Station for their work in maintaining the plots and infrastructure; J. Y. Jia, M. X. Xie, A. Yoder, G. Ni and other lab members for the data collection and molecular library preparation assistance. G.W. was supported by the National Natural Science Foundation of China (32201330), National Key R&D Program of China (2021YFD1900901, 2022YFD1900102), and J.D.B. was supported by National Science Foundation grants DEB-1738041, BII-2120153, OIA-1656006, USDA grant 1022923 and 1030634.

## Author contributions

G.W. and J.D.B. designed this project. J.D.B. designed the field experiment. G.W., H.M.B., L.Y.P. and P.A.S. performed the field work, and G.W. performed the greenhouse experiments. G.W. and H.M.B. did DNA extraction and sequencing processing work. L.Y.P. did the above ground biomass and complementarity calculations. G.W., J.D.B., L.Y.P., F.Z. and J.Z. did the data analysis. M.B.E. analyzed the analytical model and copulas with input from G.W. and J.D.B. G.W. wrote the first draft and all authors contributed to reviewing and editing.

## Competing interests

The authors declare no competing interests.
