## [Peer Review File · Nature Communications]

Reviewers' Comments:

Reviewer #1:

Remarks to the Author:

In the manuscript "Dilution of specialist pathogens drives feedback and yield advantage in plant mixtures" the authors aimed to test how plant soil feedbacks contribute to overyielding in plant species mixtures of increasing plant species richness. The authors used highly complementing methods in which they combined a field experiment with a greenhouse experiment and a plant soil feedback modelling exercise. Overall, the authors showed that in prairie grassland increased plant productivity with increasing plant species richness can, at least in part, be explained by increased dissimilarity of the soil microbial communities and reduced negative plant soil feedback.

The strong aspects of the work are the combination of the complementing methods such that longer-term observations in the field are mechanistically linked to plant soil feedback effects tested on young plants during a short greenhouse experiment, and in which the modelling exercise provides a quantitative approach to couple pairwise interaction strengths to expected dilution of pathogens in species mixtures and overyielding.

The less strong aspects comprise two main comments for which it would be nice to see info on in the revision, at least in the discussion but preferentially also in the results as I would expect that the authors must have some info on this. The first aspect I found missing is quantitative info of the plant species specific pathogens. Which pathogen was specific for which plant species and how does its abundance decline in absolute terms in species mixtures? The second aspect I found missing is the role of abiotic soil parameters and that of resource partitioning. Both mechanisms of release of negative biotic plant soil feedback and resource partitioning can cooccur as mechanisms underlying overyielding with increasing plant diversity (e.g. see review Barry et al. (2019) *The Future of Complementarity, Trends in Ecology and Evolution*). I expected some tests/discussion on this latter mechanism too as despite significant effects of plant soil feedback via soil biota there is still a considerably large amount of unexplained variation in the relation between plant species richness and productivity.

In the current manuscript the quantification of pathogen dilution is based on the dissimilarity between the microbial community compositions of the different plant species. I expected to also see a more quantitative approach to the plant species specific pathogens for example via qPCR of the targeted pathogens in the soil and roots. If qPCR was not performed that at least showing stacked bars of the relative abundances of the different plant species specific plant pathogens per plant species next to each other (for which one would expect that each plant species would then show a dominance of a different plant pathogen) and their relative distribution in species mixtures of the different plant species (for which one then would expect a more even distribution) would be enlightening.

Detailed comments

Methods line 29-30: how was the soil sampled from the 240 plots? Core depth, diameter, nr,

volume soil sample? How were the roots sampled for the extraction of the root microbes?

Soil abiotic info, pH, nutrient availability, of the soil in the different plant communities in the field experiment?

Microbial communities: info on their taxonomic identity?

Quantitative info on the microbial abundances?

Reviewer #2:

Remarks to the Author:

The study investigates the relationship between plant diversity and productivity in the presence of pathogens. Through field experiments, greenhouse assays, and feedback modeling, the study finds that pathogen dilution drives productivity benefits from diversity, suggesting that the loss of diversity could result in reduced productivity due to pathogen-mediated feedback.

The results are interesting and novel, and the experimental design is solid. My main concern

however is on the reliability of the conclusions. While the results show that pathogens have somewhat of a role in explaining productivity patterns (Extended Data Table 2), the approach used here, in my opinion, does not allow us to exhaustively infer that pathogen dilution is the MAIN microbial-related driving factor, as otherwise framed in the text. Indeed, the relative importance of pathogens was tested only against the total bacterial and fungal communities, and AM fungi. However, other trophic guilds and functional groups (e.g., fungal saprotrophs or N-fixing bacteria, which contribute to increased nutrient availability) could be important in defining the observed patterns. I believe those should be also tested to provide conclusive evidence on the driving role of pathogens in plant productivity. Additional specific comments on points that need clarification/revision are below:

Abstract, L28 – not clear what the authors mean by “overyielding measured by complementarity”. Overall, it is not clear in the abstract what the authors mean by complementarity

P4, L88 – define complementarity here

L110-113: see my general comment above

L127-129: as above, while these associations are significant, pathogens explain a little proportion of the PSF. Indeed, other additional possible explanatory factors have not been taken into consideration – for example, changes in nutrient availability or other edaphic properties, or changes in the proportion of other important functional groups (e.g., saprotrophs or N fixing bacteria). Note that tools such as FungalTraits allow for the partitioning of the fungal communities into different lifestyles.

Methods:

P21, L26-28: why was the rainfall treatment not considered in the second growing season? If I understand correctly, there were 4 months of rainfall manipulation applied before the second harvest. Have you checked that the water treatment did not impact the results? Water is very important for pathogen establishment, especially Oomycetes, and I think the lack of rainfall treatment effect needs to be statistically assessed before ruling it out.

P22 L37: I am not sure you can consider the relationship between biomass and cover as a strong one: the R-value of the correlation coefficient is 0.76, corresponding to an R-squared value of ~ 0.57 . This means that cover explains about half of the biomass variability.. is that sufficient to use cover as a proxy for biomass/productivity?

L69-94 of the methods (DNA extraction and sequencing processing) – this part is also in the supplementary? Please check. Note that the FungalTraits database is now available for assigning fungal guilds. Notably, FungalTraits incorporates FunGuild information, which has been subsequently revised by expert fungal taxonomists. As such, a justification for using FunGuild in your case needs to be provided.

See: "FungalTraits: a user-friendly traits database of fungi and fungus-like stramenopiles"

L91-93 what threshold was used for rarification?

Fig. 2. The heatmaps are almost impossible to read as the font size is very small and the resolution bad, as such I cannot really interpret those figures with confidence. Also, the Y-axis labeling is missing in figure e. In the caption, it says that e and f represent regressions. In this case, the r should be r-squared? check other figures too

Supplementary information:

L 36: “Nurse plants were inoculated with native soil microbes and then grown for two months prior to planting in the field to ensure that the microbes had a chance to establish on their host plants.” To be reproducible and assessable, this section needs more details on what soil microbes were added and how.

L55 “winder” is “winter?”

L126: which version of Qiime2 was used?

L163: spell out RYT

Reviewer #3:

Remarks to the Author:

The specific approach they used to model the host-pathogen dynamics actually uses a constant parameter to represent the pathogen effect on a particular plant relative to the pathogen effect on its copetitor (their sigma). It seems like a perfectly good way to do what they want to do. Given

that assumption then their approach is standard simulation, looking at the eigenvalues to establish stability. I'm not familiar with the general modeling approach, but their explanation is convincing to me and I see nothing wrong with it.

Regarding the random forest regression, I could not find any reference to this in the manuscript. I'm sorry but maybe my brain is just too fogged up these days, but I'm drawing a blank on that particular question.

I read the whole manuscript and it seems to me a major contribution to our understanding of the dynamics of biodiversity. I'm not familiar with all the field and lab techniques they used, but as far as I can tell this is a major contribution to this confusing literature. Very convincing on the role of pathogens in the determination of over yielding and thus the diversity productivity general pattern.

Dear Reviewers,

We greatly appreciate your constructive comments and suggestions that have helped us to significantly improve the manuscript for this revision. We have performed additional analysis that has further validated and strengthened all of our major conclusions in the manuscript. Below we provide comments to each of the reviewer's suggestions or concerns.

Reviewer #1 (Remarks to the Author):

In the manuscript "Dilution of specialist pathogens drives feedback and yield advantage in plant mixtures" the authors aimed to test how plant soil feedbacks contribute to overyielding in plant species mixtures of increasing plant species richness. The authors used highly complementing methods in which they combined a field experiment with a greenhouse experiment and a plant soil feedback modelling exercise. Overall, the authors showed that in prairie grassland increased plant productivity with increasing plant species richness can, at least in part, be explained by increased dissimilarity of the soil microbial communities and reduced negative plant soil feedback. The strong aspects of the work are the combination of the complementing methods such that longer-term observations in the field are mechanistically linked to plant soil feedback effects tested on young plants during a short greenhouse experiment, and in which the modelling exercise provides a quantitative approach to couple pairwise interaction strengths to expected dilution of pathogens in species mixtures and overyielding.

Response: we thank the reviewer for the positive comments on our design and importance on the manuscript.

The less strong aspects comprise two main comments for which it would be nice to see info on in the revision, at least in the discussion but preferentially also in the results as I would expect that the authors must have some info on this. The first aspect I found missing is quantitative info of the plant species specific pathogens. Which pathogen was specific for which plant species and how does its abundance decline in absolute terms in species mixtures?

Response: We thank the reviewer making this suggestion and we agree that this is an interesting question. Quantifying soil pathogens is challenging, particularly with the high diversity of soil pathogens that we have found in our plots. We do not have apparent pathogen symptoms in the roots that can be quantified. Furthermore, we would be unable to collect an adequate volume of roots from our plots without negatively impacting the longevity the perennial plants we have established in our plots and wish to continue studying. Given this constraint, the best method to obtain the quantitative information is using the qPCR. However, this requires specific primers for each particular pathogen. Given that we have many possible pathogens that could be important, it is not feasible to scan for specific pathogens across the eighteen plant species in our study. We talked about this in the introduction section (lines 55-59: *Testing the importance of pathogen dilution in plant communities is challenging given the high taxonomic diversity of pathogens and difficulties measuring their abundance. Conclusions about pathogen importance, for example, may depend on which of many pathogens in a particular system one selects; and even if influential pathogens are chosen, their effects may depend on interactions with ones that were omitted*). Therefore, in the revised manuscript, we address this comment of the reviewer, but at a higher taxonomic level.

Specifically, as suggested by the reviewer below, we calculated the relative abundance of the

fungal pathogens (classified by the Fungaltraits database) for each plant species and displayed these abundances in the stacked bar charts (Fig. 1 below, Supplementary Fig. 3 in the new submitted manuscript). The most abundant species are assumed to be the host specific pathogens and we observed the most abundant fungal pathogens differed for most plant species, which supports the host specific pathogen effect. To test whether pathogen abundance declines in diverse mixtures, we also calculated the change in relative abundance of the most abundant pathogen in different plant richness (Figs. 2-4 below, Supplementary Figs. 6-8 in the new version). We note that this is an imperfect measure as it is possible, if not likely, that the overall density of pathogenic fungi is changing across plots (e.g. it could decline with diversity), which would not be detected with amplicon sequencing. Nevertheless, using this approach, we found that the relative abundance of specific pathogens declined with diversity in four examples (no examples showed significant increases). *Cladosporium* in SCHSCO (Fig. 2f), *Fusarium* in EUPALT (Fig. 4c), *Didymella* in LIAPYC (Fig. 4e) and *Coniochaeta* in SILINT (Fig. 4f) were significantly reduced in monoculture compared with mixtures, in part supporting the pathogen dilution hypothesis.

We added these results in the new version of the manuscript (lines 142-144, 170-173 in the main text and Supplementary Results and Discussion).

Fig. 1 The relative abundance of soil fungal pathogens at the genus level for each plant species

Fig. 2 The relative abundance changes of the most abundant pathogen species at the genus level for each grass species. The most abundant genus is shown at the top left or top right corner of the plot.

Fig. 3 The relative abundance changes of the most abundant pathogen species at the genera level for each legume species. The most abundant genus is shown at the top left or top right corner of the plot.

Fig. 4 The relative abundance changes of the most abundant pathogen species at the genera level for each composite species. The most abundant genus is shown at the top left or top right corner of the plot.

The second aspect I found missing is the role of abiotic soil parameters and that of resource partitioning. Both mechanisms of release of negative biotic plant soil feedback and resource partitioning can cooccur as mechanisms underlyingoveryielding with increasing plant diversity (e.g. see review Barry et al. (2019) The Future of Complementarity, Trends in Ecology and Evolution). I expected some tests/discussion on this latter mechanism too as despite significant effects of plant soil feedback via soil biota there is still a considerably large amount of unexplained variation in the relation between plant species richness and productivity.

Response: We agree with the reviewer that resources partitioning is an important mechanism potentially generating the biodiversity-productivity relationship, and that it may well help explain variation in complementarity in our experiment. While the novelty of our study is the strong evidence we find supporting the idea that the dilution of soil pathogens are drivers of productivity responses to diversity, it would be interesting to explore whether there is evidence of resource partitioning. However, we do have two sources of information that fit into the present study. The first is the effect of legumes and the second comes from analyses of soil resources in plots (described below).

Previous studies have found that combining N-fixing legumes with grasses can play important roles in driving the resources partitioning, as legumes could fix N_2 from the air and alleviate competition of soil N with non-fixing plants (Tilman et al. 2001). We evaluated whether we could see evidences of this mechanism in our data by recalculating the relationships between predicted PSF effect, predicted pathogen dilution and complementarity effect and also relative yield with N-fixing legumes either be included or excluded (Fig. 5 below, Supplementary Fig. 5 in the new version). We found that relationships were similar for “withlegume” (mixture plots with legumes) and “nolegume” (mixture plots without legumes) groups. Specifically, we found that regression slopes were not statistically significant as the interaction between predicted PSF or pathogen dilution with legume inclusion/exclusion were not significant (Table 1 below, Supplementary Table 7 in the new

version). These results indicate that biotic PSF effects play an important role in explaining the plant biodiversity effect, even in plant mixtures with legumes. While not significantly different, we can observe that the biotic PSF effect had a lower explained variance (higher determination coefficient) for complementarity oroveryielding effect in withlegume than nolegume groups, perhaps consistent with resources partitioning making a difference. This possibility will have to be tested in future studies, which we identify in the revised manuscript.

We added the relevant results and discussion in lines 166-170 in the main text and Supplementary Results and Discussion in the new version.

Tilman, D., Reich, P. B., Knops, J., Wedin, D., Mielke, T., & Lehman, C. (2001). Diversity and productivity in a long-term grassland experiment. Science, 294(5543), 843-845.

Fig. 5 The relationship between predicted plant-soil feedback (PSF), pathogen dilution effect and complementarity effect (a,b) and relative yield total (c,d) inclusion or exclusion of N-fixing legumes. The presence or absence of N-fixing legumes is indicated in blue (withlegume) and red (nolegume), respectively.

Table 1 Significant difference tests on the regression models between complementarity effect and relative yield total and predicted PSF and pathogen dilution effect under withlegume and nolegume treatments.

	Complementarity effect				Relative yield total			
	Estimate	SE (standard error)	t value	P	Estimate	SE (standard error)	t value	P
Intercept	45.46	17.97	2.53	0.0123	1.20	0.09	13.21	< 0.001
Predicted PSF	-162.51	43.96	-3.70	<0.001	-0.89	0.22	-4.00	< 0.001
Legume	35.96	24.99	1.44	0.15	0.14	0.13	1.12	0.26
Predicted PSF × Legume	53.68	60.1	0.89	0.37	0.35	0.30	1.16	0.25
Intercept	52.69	16.01	3.29	0.0012	1.25	0.08	15.28	< 0.001
Predicted pathogen dilution (PPD)	208.73	54.46	3.83	<0.001	1.10	0.28	3.94	< 0.001
Legume	30.12	22.14	1.36	0.18	0.12	0.11	1.08	0.28
PPD × Legume	-59.17	73.17	-0.81	0.42	-0.45	0.37	-1.20	0.23

In the current manuscript the quantification of pathogen dilution is based on the dissimilarity between the microbial community compositions of the different plant species. I expected to also see a more quantitative approach to the plant species specific pathogens for example via qPCR of the targeted pathogens in the soil and roots. If qPCR was not performed that at least showing stacked bars of the relative abundances of the different plant species specific plant pathogens per plant species next to each other (for which one would expect that each plant species would then show a dominance of a different plant pathogen) and their relative distribution in species mixtures of the different plant species (for which one then would expect a more even distribution) would be enlightening.

Response: We thank the reviewer for this suggestion. As shown in our above response, we have incorporated this suggestion in the revised version of the manuscript.

Detailed comments

Methods line 29-30: how was the soil sampled from the 240 plots? Core depth, diameter, nr, volume soil sample? How were the roots sampled for the extraction of the root microbes?

Response: We moved some of the information previously presented in the Supplementary Information to the M&M section in the revised manuscript. Hence, the information requested by the reviewer is now present in the Methods section (lines 275-286 in the new version):

Soils were collected from each plot in September 2018, four months after planting, and then paired plots of matched plant composition were pooled across paired rainfall exclusion shelters. A total of six 20 cm soil cores with 1.9 cm diameter (approximately 340 ml) were taken from each plot, sealed in polyethylene bags and kept on ice and then transferred to the lab for storage. The coring devices were sanitized between different treatments to minimize contamination. The soil cores were taken close to planted species to be more representative of plant-affected soil microbes and to ensure we were able to collect root samples. The soil samples were separated into two subsamples. One subsample (approximately 50 ml) was passed through a 4-mm mesh to separate roots, and then both soil and sterile water washed root samples were stored in a freezer at -20°C for DNA extraction and sequencing. The remaining soil was stored at 4°C until it was used for the subsequent plant-soil feedback experiment.

Soil abiotic info, pH, nutrient availability, of the soil in the different plant communities in the field experiment?

Response: We measured the soil pH, organic matter, total carbon and nutrients (total N and P) from the soil collected in the September 2018 sampling, but no significant differences were found between plots with different levels of plant richness for all five parameters (Fig. 6). We understand that the reviewer may want to link the changes of soil nutrients uptake with resource partitioning in different plant communities. However, over the timescale of this experiment, we did not expect to see changes in SOM, N, P, or pH. Changes in soil C, N, P, and pH are often detectable on multidecadal, rather than annual timescales (McLauchlan et al. 2006; Rosenzweig et al. 2016; Tweedie et al. 2021). While plant diversity can accelerate the development of those resource pools (Lange et al. 2015; Rosenzweig et al. 2016; Tilman and Furey 2021; Yang et al. 2019), those changes are considered rapid when they occur over a decade, rather than multiple decades. As we acknowledge above and in the last paragraph of our manuscript (lines 237-243), though soil abiotic properties did not change significantly, resource partitioning effects may operate in concert with pathogen dilution. However, in this manuscript, we instead thoroughly explore the underlying

mechanism involving biotic PSF, which are less well documented in the literature. Given biotic facilitation has not been demonstrated in the context of grassland diversity manipulation experiments, we felt it was important to thoroughly present evidence of the mechanism using complementary approaches, field experiment, greenhouse assays, and feedback modeling, which warranted the thorough and focused description we present in this manuscript.

Lange, M., Eisenhauer, N., Sierra, C.A., Bessler, H., Engels, C., Griffiths, R.I., Mellado-Vázquez, P.G., Malik, A.A., Roy, J., Scheu, S. and Steinbeiss, S. (2015). Plant diversity increases soil microbial activity and soil carbon storage. *Nature communications*, 6(1), 6707.

McLauchlan, K.K., Hobbie, S.E. and Post, W.M. (2006). Conversion from agriculture to grassland builds soil organic matter on decadal timescales. *Ecological applications*, 16(1), 143-153.

Tweedie, A., Haygarth, P.M., Edwards, A., Lilly, A., Baggaley, N. and Stutter, M. (2021). Soil phosphorus over a period of agricultural change in Scotland. *European Journal of Soil Science*, 72(6), 2457-2476.

Rosenzweig, S.T., Carson, M.A., Baer, S.G. and Blair, J.M. (2016). Changes in soil properties, microbial biomass, and fluxes of C and N in soil following post-agricultural grassland restoration. *Applied Soil Ecology*, 100, 186-194.

Yang, Y., Tilman, D., Furey, G. and Lehman, C., 2019. Soil carbon sequestration accelerated by restoration of grassland biodiversity. *Nature communications*, 10(1), p.718.

Furey, G.N. and Tilman, D. (2021). Plant biodiversity and the regeneration of soil fertility. *Proceedings of the National Academy of Sciences*, 118(49), e2111321118.

Fig. 6 Soil parameters in different plant richness.

Microbial communities: info on their taxonomic identity?

Quantitative info on the microbial abundances?

Response: As noted in our response above, we followed the suggestion of the reviewer and

calculated the relative abundance of the fungal pathogens (using a classification based on the Fungaltraits database as suggested by Reviewer 2) as found in each plant species monoculture and displayed as the stacked bar plot (Fig. 1, Supplementary Fig. 3 in the new version). In order to test whether pathogens' abundance declined in species mixtures, we also calculated the relative changes in abundance of the most abundant pathogen across levels of plant richness. While this method may be viewed as conservative, because relative proportion may not map on directly to real densities, it did yield interesting results for several plant species (Supplementary Figs. 6-8).

Reviewer #2 (Remarks to the Author):

The study investigates the relationship between plant diversity and productivity in the presence of pathogens. Through field experiments, greenhouse assays, and feedback modeling, the study finds that pathogen dilution drives productivity benefits from diversity, suggesting that the loss of diversity could result in reduced productivity due to pathogen-mediated feedback. The results are interesting and novel, and the experimental design is solid.

Response: we thank the reviewer for the positive comments on our design and novelty on the manuscript.

My main concern however is on the reliability of the conclusions. While the results show that pathogens have somewhat of a role in explaining productivity patterns (Extended Data Table 2), the approach used here, in my opinion, does not allow us to exhaustively infer that pathogen dilution is the MAIN microbial-related driving factor, as otherwise framed in the text. Indeed, the relative importance of pathogens was tested only against the total bacterial and fungal communities, and AM fungi. However, other trophic guilds and functional groups (e.g., fungal saprotrophs or N-fixing bacteria, which contribute to increased nutrient availability) could be important in defining the observed patterns. I believe those should be also tested to provide conclusive evidence on the driving role of pathogens in plant productivity.

Response: As noted by the reviewer in the comment below, tools such as FungalTraits allow for the partitioning of the fungal communities into different lifestyles. Following the suggestion of the reviewer, we reclassified the fungal community using this FungalTraits database, and separated bacteria into rhizobia and non-rhizobia. In total, 212 Amplicon Sequence Variants (ASVs) were categorized as plant pathogens with the Fungaltraits database. In contrast, when using the Funguild database, which we used to prepare the previous version of the manuscript, we only found 164 ASVs with the confidence ranking of “Probable” and “Highly Probable” of being pathogens. Comparing the classifications of both databases 112 ASVs were classified as pathogens in both the Funguild and FungalTraits databases. Therefore, we believe the Fungaltraits database provided a more comprehensive assessment of fungal pathogens, and we thank the reviewer for this suggestion.

When calculating the predicted pairwise feedback, we used 12 potential microbial predictors (rhizobia, non-rhizobia bacteria, AMF, fungal pathogen, fungal saprotroph and oomycete for both soil and root). However, for soil rhizobia, only 28 ASVs were identified, leading to many missing values in calculations of dissimilarities and could therefore not be utilized as predictors. Therefore, we constructed models using the other 11 microbial predictors. We found that soil fungal pathogen, soil oomycete and root pathogen dissimilarities were the three most important predictors and their effects on predicting the PSF were significant (Supplementary Table 3). The regressions between soil fungal pathogen and oomycete dissimilarities and measured pairwise PSFs were also significant and indicate the importance of pathogens in driving negative PSFs (Fig. 2e,f and Supplementary Table 2). This is qualitatively consistent with our previous results from Funguild (which showed soil fungal pathogens and soil oomycetes alone as the best predictors) in the initial submission.

We added the relevant information in the Methods section in the new version (lines 337-342, lines 375-395):

(lines 337-342 in the Methods section) *Functional groups within the fungal community were*

categorised based on the FungalTraits database⁵⁹. The amplicon sequence variants (ASVs) categorised as “plant_pathogen” in the “primary_lifestyle” or “secondary_lifestyle” were used as putative pathogens, and “soil_saprotroph”, “litter_saprotroph”, “wood_saprotroph” and “unspecified_saprotroph” in the “primary_lifestyle” were used as saprobes for subsequent analyses.

(lines 378-398 in the Methods section) We then considered two different approaches (linear model and random forest) to predict pairwise feedback using 12 potential microbial predictors (rhizobia, non-rhizobia bacteria, AMF, fungal pathogen, fungal saprobe and oomycetes for both soil and root). However, for soil rhizobia, only 28 ASVs were identified and there were many missing values when calculating the dissimilarity and could not be used in later modeling running. Therefore, we removed the soil rhizobia when running the models while keeping the other 11 microbial predictors. Linear and random forest models⁶⁵ were compared using the *lm* and *randomForest* base R functions, and we assessed the significance of fitted random forest models using the *A3* package⁶⁶. We found that the linear model explained a larger proportion of variance ($R_{adj2} = 0.137$) than the random forest model ($R_{adj2} = 0.106$). Then, we performed a model selection process for feedback effects, using the *glmulti* package in R⁶⁷ to generate a suite of candidate models and sample size corrected Akaike Information Criterion (AICc)⁶⁸ to distinguish between candidate models. As $\Delta AICc$ between models were statistically indistinguishable (≤ 2), we opted to perform model averaging to represent the importance of parameters across all candidate models⁶⁹. Parameter weights were estimated as the sum of Akaike weights across all the models⁷⁰ and are considered a measure of the overall support for each predictor. We set a lower limit of 0.7 to differentiate between the important and unimportant predictors. We found that soil fungal pathogen, soil oomycete and root pathogen dissimilarities were the three most important predictors and their effects on predicting the PSF were significant (Supplementary Table 3).

Additional specific comments on points that need clarification/revision are below:

Abstract, L28 – not clear what the authors mean by “overyielding measured by complementarity”.

Overall, it is not clear in the abstract what the authors mean by complementarity

Response: The plant biodiversity effect can be constructively divided into two additive components: (i) the selection effect, which is the portion of increase in production with increasing diversity that results from the increased probability of including in the mixture the species with highest individual performance; and (ii) the complementarity effect, which is the portion of the change in production with increasing diversity that results from the niche differentiation among species in a mixture (Loreau & Hector 2001). In both diversity–productivity or diversity–disease relationships, we can assess whether the effect of diversity on the ecosystem service depends on the identity of species added to the community because particular species play a disproportionate role in improving ecosystem function (selection effects) or is a consequence of the number of species per se (complementarity) (Collins et al. 2020). Relative yield total (RYT) was used to compare the productivity of sole and multiple cropping systems, namely overyielding. The mixture yields of a component crop expressed as a portion of its yields as a sole crop from the same replacement series is the relative yield of crop, and sum of the relative yields of component crop is called RYT. In this study, we found a significant positive relationship between RYT and complementarity effect rather than a selection effect (Supplementary Fig. 4), suggesting that the overyielding effect was mainly

determined by the complementarity effect. We understand that our previous formulation in the Abstract might have been unclear without the additional context described above. Hence we have reformulated the sentence as:

Lines (29-31) “*We found the rapid accumulation of specialist pathogens in monocultures decreased yields of their host plants and pathogen dilution predicted field plant community productivity gains from diversity.*”

Loreau, M. & Hector, A. *Partitioning selection and complementarity in biodiversity experiments. Nature* 412, 72-76 (2001).

Collins, C.D. et al. (2020) *Community context for mechanisms of disease dilution: insights from linking epidemiology and plant–soil feedback theory. Annals of the New York Academy of Sciences* 1469 (1), 65-85.

P4, L88 – define complementarity here

Response: As suggested by the reviewer, we defined the complementarity here as “*a measure of overyielding due to interactions between species*” (line 90), in this case due to the density of host-specific pathogen being diluted by the presence of other species.

L110-113: see my general comment above

Response: Following the suggestion of the reviewer, we reclassified the fungal community using the FungalTraits database and reconducted the models. We found that even with more detailed consideration of soil microbial community components (11 predictors), we find that only pathogenic groups significantly explain observed PSFs, indicating the importance of pathogens in driving negative PSFs.

L127-129: as above, while these associations are significant, pathogens explain a little proportion of the PSF. Indeed, other additional possible explanatory factors have not been taken into consideration – for example, changes in nutrient availability or other edaphic properties, or changes in the proportion of other important functional groups (e.g., saprotrophs or N fixing bacteria). Note that tools such as FungalTraits allow for the partitioning of the fungal communities into different lifestyles.

Response: Please see the response above. We also want to address that in this study much attention has been paid on the underlying mechanism of the biotic PSF and we demonstrated its importance in driving plant community complementarity and overyielding effects. However, we agree with the reviewers that other mechanisms such as resources partitioning cannot be completely ruled out and need further examination in the future. However, as we responded to Reviewer 1, over the timescale of this experiment, we did not expect to see changes of soil nutrient availability (see Figure 6 above). Changes in soil C, N, P, and pH are often detectable on multidecadal, rather than annual timescales (McLauchlan et al. 2006; Rosenzweig et al. 2016; Tweedie et al. 2021). While plant diversity can accelerate the development of those resource pools (Lange et al. 2015; Rosenzweig et al. 2016; Tilman and Furey 2021; Yang et al. 2019), those changes are considered rapid when they occur over

a decade, rather than multiple decades. As we acknowledge above and in the last paragraph of our manuscript (lines 240-246), though soil abiotic properties did not change significantly, resource partitioning effects may operate in concert with pathogen dilution. However, in this manuscript, we instead thoroughly explore the underlying mechanism involving biotic PSF, which are less well documented in the literature. Given biotic facilitation has not been demonstrated in the context of grassland diversity manipulation experiments, we felt it was important to thoroughly present evidence of the mechanism using complementary approaches, field experiment, greenhouse assays, and feedback modeling, which warranted the thorough and focused description we present in this manuscript. We note that the experimental protocol for the greenhouse feedback test used small amounts of soil as inocula in order to minimize abiotic differences. The fact that pathogen differentiation predicts soil feedback affirms that the biotic differences drive a portion of the feedback. While it remains possible that other components of the microbiome contributed to the feedback, we do not have evidence of this within our experiments.

Furthermore, N-fixing legumes may facilitate resource partitioning, as legumes may form symbiotic relationships with rhizobia capable of fixing N_2 from atmospheric sources. This may alleviate competition for soil N with non N-fixing plants, so we recalculated the relationships between predicted PSF effect, predicted pathogen dilution and complementarity effect and also relative yield total inclusion or exclusion of N-fixing legumes (Fig. 5, Supplementary Fig. 5 in the new version). Similar trends were found between withlegume (mixture plots with legumes) and nolegume (mixture plots without legumes) groups, and the regression slopes were not statistically different as the interaction between Predicted PSF or pathogen dilution with legume were not significant (Table 1, Supplementary Table 7 in the new version). Our results indicate that the biotic PSF effect plays an important role in explaining the plant biodiversity effect, even in plant mixtures with legumes. However, we can see that the biotic PSF effect had a lower explained variance (higher determination coefficient) for complementarity or overyielding effect in withlegume than nolegume groups, indicating that resource partitioning may make a difference. Therefore, resource partitioning effects should be further explicitly studied for future studies. Based on these results, we added the relevant results and discussion in lines 166-170 in the main text and Supplementary Results and Discussion in the new version.

Lange, M., Eisenhauer, N., Sierra, C.A., Bessler, H., Engels, C., Griffiths, R.I., Mellado-Vázquez, P.G., Malik, A.A., Roy, J., Scheu, S. and Steinbeiss, S. (2015). Plant diversity increases soil microbial activity and soil carbon storage. *Nature communications*, 6(1), 6707.

McLauchlan, K.K., Hobbie, S.E. and Post, W.M. (2006). Conversion from agriculture to grassland builds soil organic matter on decadal timescales. *Ecological applications*, 16(1), 143-153.

Tweedie, A., Haygarth, P.M., Edwards, A., Lilly, A., Baggaley, N. and Stutter, M. (2021). Soil phosphorus over a period of agricultural change in Scotland. *European Journal of Soil Science*, 72(6), 2457-2476.

Rosenzweig, S.T., Carson, M.A., Baer, S.G. and Blair, J.M. (2016). Changes in soil properties, microbial biomass, and fluxes of C and N in soil following post-agricultural grassland restoration. *Applied Soil Ecology*, 100, 186-194.

Yang, Y., Tilman, D., Furey, G. and Lehman, C., 2019. Soil carbon sequestration accelerated by restoration of grassland biodiversity. *Nature communications*, 10(1), p.718.

Furey, G.N. and Tilman, D. (2021). Plant biodiversity and the regeneration of soil fertility.

Methods:

P21, L26-28: why was the rainfall treatment not considered in the second growing season? If I understand correctly, there were 4 months of rainfall manipulation applied before the second harvest. Have you checked that the water treatment did not impact the results? Water is very important for pathogen establishment, especially Oomycetes, and I think the lack of rainfall treatment effect needs to be statistically assessed before ruling it out.

Response: We agree with the reviewer that precipitation could be very important for pathogen establishment. Indeed, this was a major reason for initiating this manipulation. However, within the first growing season we were not able to implement the precipitation treatment in full because the shelter construction was not completed until August 2018, thus both treatments were open and receiving ambient precipitation. The summer of 2018 Kansas was experiencing a drought (Fig. 7 below) and to further drought the experiment would have likely inhibited the establishment of our plants. Since we were not able to fully implement our precipitation treatment before the September harvest, we decided to focus our limited efforts on the biodiversity effects and effects of roots versus soils for the microbial sampling at the end of the first growing season. Soil samples were pooled across paired diversity treatments within the rainfall exclusion shelters (i.e. we pooled samples from plots with the same plant composition from the 50% and 150% treatments) before analysis of microbiome composition, soil properties, and plant soil feedback. Therefore, we did not expect to see differences in microbiomes between high and low precipitation treatments within this experiment.

Beginning in spring 2019, both high and low precipitation shelters were covered, giving us the ability to effectively manipulate precipitation. After that, one paired rainfall exclusion shelter received 150% ambient growing season rainfall, while the paired rainout shelter received 50% ambient precipitation. Therefore, when using the predicted PSF and pathogen dilution effect predict the complementarity effect and RYT, we used all the 168 plant mixture plots, including both 50% and 150% water treatment. As the response to the reviewer, we further separated the 150% and 50% water treatment to look at their respective patterns. We found that the regressions showed similar trends in both 50% and 150% treatment (Fig. 8 below), and the two regression slopes were not statistically significant as the interaction between predicted PSF or pathogen dilution with water treatment were not significant (Table 2 below). These results are consistent with PSF playing important roles in explaining the plant biodiversity effect.

Fig. 7 Cumulative annual and growing season (April-Oct) precipitation measured at the University of Kansas Field Station. The first year of the experiment, 2018, is indicated by the dashed line, while the second year is indicated by the dotted line. We compare year one and two precipitation to the mean cumulative precipitation for the site, solid line, measured across 1979-2022. Cumulative precipitation is within the 90th and 10th quartile of expectations for the site when it is within the grey shaded polygon.

Fig. 8 The relationship between predicted plant-soil feedback (PSF), pathogen dilution effect and complementarity effect (a,b) and relative yield total (c,d) in 50% and 150% water treatment.

Table 2 Significant difference tests on the regression models between complementarity effect and relative yield total and predicted PSF and pathogen dilution effect under 50% and 150% water treatments.

	Complementarity effect				Relative yield total			
	Estimate	SE (standard error)	t value	P	Estimate	SE (standard error)	t value	P
Intercept	78.95	18.18	4.34	< 0.001	1.32	0.09	14.36	< 0.001
Predicted PSF	-121.27	43.39	-2.80	0.0058	-0.71	0.22	-3.26	0.0013
Water	-28.32	25.03	-1.13	0.26	-0.08	0.13	-0.67	0.50
Predicted PSF × Water	-27.18	59.93	-0.45	0.65	0.03	0.30	0.11	0.91
Intercept	81.73	15.90	5.14	< 0.001	1.35	0.08	16.66	< 0.001
Predicted pathogen dilution (PPD)	164.12	51.84	3.17	0.0018	0.89	0.26	3.38	< 0.001
Water	-25.72	22.12	-1.16	0.25	-0.07	0.11	-0.66	0.51
PPD × Water	28.30	72.48	0.39	0.70	-0.09	0.37	-0.26	0.80

P22 L37: I am not sure you can consider the relationship between biomass and cover as a strong one: the R-value of the correlation coefficient is 0.76, corresponding to an R-squared value of ~0.57. This means that cover explains about half of the biomass variability. is that sufficient to use cover as a proxy for biomass/productivity?

Response: The strength of our experimental design is high statistical power that comes from many plots. With hundreds of plots, we are able to replicate monocultures of all planted species, and we are able to equally represent each planted species in each treatment combination. However, having many plots comes with a trade-off in plot size. While our plots are of sufficient size to capture plant species interactions (Roscher et al. 2005), we cannot afford annual plot scale harvests at peak biomass, as this could damage the long-term viability of our experiment. To limit damage to the plots, we visually score cover of individual plant species and harvest approximately 1/10 of the area of the plot at peak biomass each summer. We then convert cover estimates to biomass estimates using species-specific relationships of cover to biomass obtained by regressing species biomass in the strip against plot cover for that species. Of course, there is noise in this relationship, as the biomass strip only represents a fraction of the plot and an individual strip will not represent each species in proportion to total plot cover. Since we have 240 plots, we are able to get statistically strong relationships across all plots and we believe the net result is a very good translation of cover to biomass. Importantly, the noise in the relationships of plot level percent cover with strip harvested biomass for individual species is not the noise in our estimate of plot level biomass. By scaling the biomass of individual species to measures cover of the whole plot, which we find to be highly reproducible, we feel that our estimates are a good compromise compared to strip plots alone (which is a common approach in biodiversity manipulations). The net effect will result in a reduction of the precision of the measurement of above-ground biomass compared to a complete plot harvest. Nevertheless, this reduction in precision is unbiased and will not generate false signals. Our results indicate that in spite of the unavoidable trade-offs between precision and replication, the strength of high statistical power combined with our control of species differences contributes to our being able to detect the important role of soil pathogen dilution within the second growing season (lines 453-462 in the Supplementary Methods).

Roscher, C. et al. (2005) Overyielding in experimental grassland communities—irrespective of species pool or spatial scale. Ecology Letters 8 (4), 419-429.

L69-94 of the methods (DNA extraction and sequencing processing) – this part is also in the supplementary? Please check. Note that the FungalTraits database is now available for assigning fungal guilds. Notably, FungalTraits incorporates FunGuild information, which has been subsequently revised by expert fungal taxonomists. As such, a justification for using FunGuild in your case needs to be provided.

See: "FungalTraits: a user-friendly traits database of fungi and fungus-like stramenopiles"

Response: we moved the relevant information of DNA extraction and sequencing processing into the Methods section, and we re-categorized fungal groups using the FungalTraits database. Thank you for suggesting this. We have confirmed that our story is robust to the software attributing function to fungal taxa. In fact, using FungalTraits gives stronger evidence of pathogens as the primary driver of negative feedback and stronger evidence of dilution of specialist pathogens driving complementarity. Please see more details in the responses above.

L91-93 what threshold was used for rarification?

Response: we added the threshold for each microbial community.

(lines 343-346 in the Methods section of the new version) All samples were normalized to a sequencing depth of the lowest total reads per sample (soil fungi-3880, root fungi-3943, soil bacteria-661, root bacteria-658, soil AMF-860, root AMF-358, soil oomycete-2004, root oomycete-200) prior to downstream analyses in R.

Fig. 2. The heatmaps are almost impossible to read as the font size is very small and the resolution bad, as such I cannot really interpret those figures with confidence. Also, the Y-axis labeling is missing in figure e. In the caption, it says that e and f represent regressions. In this case, the r should be r-squared? check other figures too

Response: we re-made the heatmaps and moved them into the appendix in the new version (Supplementary Fig. 2), we double-checked that the resolution of the figures is now appropriate. We added the Y-axis labeling in figure panel e (now Fig. 2c in the revised manuscript) and we changed the r value to r-squared for those regression plots. Thank you for these suggestions.

Supplementary information:

L36: "Nurse plants were inoculated with native soil microbes and then grown for two months prior to planting in the field to ensure that the microbes had a chance to establish on their host plants." To be reproducible and assessable, this section needs more details on what soil microbes were added and how.

Response: We have added information to the description as below:

"Resident soil microbes were augmented with soil microbes from native prairie in two ways. Firstly, we added 4 cm of soil from an unplowed native prairie soil from Welda, KS, which was then tilled into the resident soil to a depth of 15 cm. Secondly, as native microbes may not be resilient to tillage, we also introduced microbes by planting nurse plants. Briefly, seeds were sowed into flats with autoclaved sterile potting soil and placed in cold-moist stratification for four weeks prior to germination. When large enough, the seedlings were transplanted into groove tubes (GT51D, Stuewe and Sons, Oregon) with inoculation of 98 mL of fresh soil from unplowed prairie remnant from Welda, KS and grown in a greenhouse for 5 weeks prior to being planted. Nurse plants were inoculated with native soil microbes and then grown for two months prior to planting in the field to ensure that the microbes had a chance to establish on their host plants." (lines 333-344 in the Supplementary Methods)

L55 "winder" is "winter?"

Response: thank you and revised.

L126: which version of Qiime2 was used?

Response: Qiime2-2019.10, we added the information.

L163: spell out RYT

Response: added the full name.

Reviewer #3 (Remarks to the Author):

The specific approach they used to model the host-pathogen dynamics actually uses a constant parameter to represent the pathogen effect on a particular plant relative to the pathogen effect on its copetitor (their sigma). It seems like a perfectly good way to do what they want to do. Given that assumption then their approach is standard simulation, looking at the eigenvalues to establish stability. I'm not familiar with the general modeling approach, but their explanation is convincing to me and I see nothing wrong with it.

Response: we thank the reviewer for the positive comments on modeling work.

Regarding the random forest regression, I could not find any reference to this in the manuscript. I'm sorry but maybe my brain is just too fogged up these days, but I'm drawing a blank on that particular question.

Response: We provided the references about the random forest in the Methods section "*Linear and random forest models⁶⁵ were compared using the lm and randomForest base R functions, and we assessed the significance of fitted random forest models using the A3 package⁶⁶*". We also submitted our codes for running and comparing these two models (code availability link <https://github.com/wlzwgz/Dimension-PSF.git>).

Liaw, A. & Wiener, M. Classification and regression by randomForest. R News 2, 18-22 (2002).

I read the whole manuscript and it seems to me a major contribution to our understanding of the dynamics of biodiversity. I'm not familiar with all the field and lab techniques they used, but as far as I can tell this is a major contribution to this confusing literature. Very convincing on the role of pathogens in the determination of over yielding and thus the diversity productivity general pattern.

Response: We thank the reviewer for the positive comments on our work.

Reviewers' Comments:

Reviewer #1:

Remarks to the Author:

I was one of the referees for the original manuscript and am happy with the responses by the authors to my questions in their rebuttal and the revised version of the manuscript. I have no additional comments and look forward to see the paper published.

Reviewer #2:

Remarks to the Author:

The authors have done a good job addressing my initial concerns and I now recommend this manuscript for publication.

Reviewer #3:

None

REVIEWERS' COMMENTS

Reviewer #1 (Remarks to the Author):

I was one of the referees for the original manuscript and am happy with the responses by the authors to my questions in their rebuttal and the revised version of the manuscript.

I have no additional comments and look forward to see the paper published.

Reviewer #2 (Remarks to the Author):

The authors have done a good job addressing my initial concerns and I now recommend this manuscript for publication.

Reviewer #3 (Remarks to the Author):

No remarks.

We would like to thank all of three reviewers for taking the necessary time and effort to review the manuscript. We sincerely appreciate all of the valuable comments and suggestions, which helped a lot to improve the quality of the manuscript.